# Effects of Black Jade on Osteogenic Differentiation of Adipose Derived Stem Cells under Benzopyrene

Yoonjin Park [1,†], Gyeong Hee Shin [1,†], Gyo Sik Jin [2], Sungbae Jin [3], Boyong Kim [1,4,5,*] and Seung Gwan Lee [1,*]

[1] Department of Clinical Laboratory Sciences, College of Health Science, Korea University, Seoul 02841, Korea; pyoonjin@naver.com (Y.P.); 87075@korea.ac.kr (G.H.S.)
[2] Dr. JADE, 1-26, Baekbeom-ro 16-gil, Mapo-gu, Seoul 04149, Korea; foc2233@naver.com
[3] Department of Philosophy of Science, SunMoon University, Asan-si 336-708, Korea; krantzc4@gmail.com
[4] Life Together, 13, Gongdan-ro, Chuncheon-si 24232, Korea
[5] Mitosbio, 13, Gongdan-ro, Chuncheon-si 24232, Korea
* Correspondence: erythro74@korea.ac.kr (B.K.); seunggwan@korea.ac.kr (S.G.L.);
  Tel.: +82-10-9105-1435 (B.K.); +82-10-9913-0147 (S.G.L.)
† These authors contributed equally to this work.

**Abstract:** Jade, a popular gemstone symbolizing beauty, grace, and longevity, is known to improve blood circulation; however, scientific research evidence is still lacking. The effect of black jade extract on the expression levels of apoptotic and osteogenic genes was validated using qPCR and flow cytometry. In combination with the use of a fluorescence microscope, osteogenic differentiation and the stained osteocytes count were analyzed. Under the pressure of benzo(a)pyrene, dermal cell apoptosis was accelerated and the osteogenic differentiation of adipose-derived stem cells (ASCs) was suppressed; but black jade extract counteracted the effects. Through an anti-apoptotic mechanism, the extract suppressed the expression of apoptotic proteins Bax and cytochrome C to 9 and 4.8 times, respectively, compared to that in dermal cells exposed to benzo(a)pyrene. During osteogenic differentiation of ASCs, the extract enhanced their differentiation despite being exposed to benzo(a)pyrene, and the relative levels of the osteoblast differentiation markers osteoponin, osteocalcin, and sclerostin were 1.87, 2.54, and 3.9 times higher, respectively, than those in the conditioned medium by benzo(a)pyrene. These effects of the extract indicate that black jade extract is very useful when applied as a functional biomaterial.

**Keywords:** osteogenesis; black jade; adipose-derived stem cell; fine particles; osteocyte

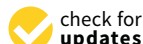



## 1. Introduction

Jade is a popular gemstone known for its discrete and luster characteristics expressed in green, white, black, gray, orange, violet, and yellow [1–3]. The gemstone is found in many countries, and its mineral composition is determined by the contents of nephrite $(Ca_2(Mg,Fe)_5Si_8O_{22}(OH)_2Si_2O_6)$ and jadeite $(Na(Al,Fe)Si_2O_6)$ [1,2]. Compared to other gemstones, the oxide composition in black jade has a 105% difference in jadeite, and black jade from Indonesia, in particular, contains high oxide compounds such as $SiO_2$ (39.6%) and $Fe_2O_3$ (35%) [2].

Benzo(a)pyrene, a polycyclic aromatic hydrocarbon, is a byproduct formed during incomplete combustion of organic matter such as meat grilling and tobacco smoking, and is released as fine particles [4]. With increasing atmospheric fine particles, a high concentration of benzopyrene has led to increased thermal stress, causing a high prevalence of dermal inflammatory diseases including atopy, acne, eczema, perivascular dermatitis, nodular, and diffuse dermatitis [5]. Exposure to benzo(a)pyrene induces DNA damage in various cells, leading to the overexpression of p53 and p21 proteins and ultimately, reduced cell viability and increasing cellular apoptosis [6].

Mesenchymal stem cells (MSCs) have been widely studied for therapeutic applications in protective and regenerative medicine [7]. Although bone-marrow-derived MSCs

(BMSCs) are very useful cells in osteogenic application [7], from the viewpoint of functional application in cosmetics, osteogenic cells derived from adipocyte-derived stem cells (ASCs) are more suitable. ASCs are differentiated from various lineages including adipocyte, chondrocyte, neuron, myocyte, and osteocyte [7–10]. In the skin, ASCs are localized to subcutaneous fat tissue and connected to other dermal cells [11]. Unlike the inhibition of osteogenic differentiation, fibroblast growth factor-2 (FGF-2) activates adipogenic and chondrogenic differentiations in ASCs [12]. In addition, matrix metalloproteinase (MMP) release is increased during damage to the connective tissue [13] and osteogenic differentiation is inhibited by MMPs in ASCs [14].

During the osteogenesis of mesenchymal stem cells, the levels of expression of various genes change with development stages. In pre-osteoblasts, levels of RUNX2 and DLX5 are increased, osteoponin and osteoponin are expressed in osteoblasts and SOST-sclerostin is expressed in osteocytes [15,16]. In addition, p53 down-regulates osterix and RUNX2 in ASCs [15].

This study aimed to investigate the protective effect of black jade extract on dermal cells under benzo(a)pyrene stress and the altered osteogenic differentiation in ASCs using activated fibroblast cells. In addition, this research documented the interaction between ASCs and dermal cells exposed to benzo(a)pyrene.

## 2. Materials and Methods

### 2.1. Cell Culture and Black Jade Extract

Human fibroblast cells purchased from Korea Cell Line Bank, Seoul, Korea, were cultured in DMEM (Dulbecco's Modified Eagle Medium) in high glucose (Invitrogen, Carlsbad, CA, USA) and supplemented with 10% FBS (sigma, St. Louis, MO, USA)and 100 U/mL of penicillin (Invitrogen), at 37 °C with 5% humidified $CO_2$. Unlike the control (black jade), which was crushed using a general grinder (RT-04SF, Rong Tsong Precision Tech Co, Taiwan, China) without heating, black jade was exposed to a high temperature (1350 °C) and the heated black jade was crushed using a micro-grinder (S3000, Korea Powder Solution. Co., LTD, Seoul, Korea) for two cycles (Figure 1). This black jade was supplied by Dr. JADE Co., Ltd. (Seoul, Korea) and Life Together (Chuncheon, Korea). After hydrolysis, the black jade extract (1000 μg/mL) was used to treat fibroblast cells for 3 days. These fibroblast cells were subsequently exposed to 15 μM/mL of benzo(a)pyrene for 1 day and the supernatants (300 μL/mL) were used to treat ASCs. Before treatment using the supernatants, ASCs (Thermo Fisher Scientific, Waltham, MA, USA) were cultured in a MesenPRO RS™ Basal Medium, Gibco, (Thermo Fisher Scientific, Waltham, MA, USA) with growth supplement (MesenPRO RS™ Growth Supplement, Thermo Fisher Scientific, Waltham, MA, USA).

To estimate the protective function for osteogenic differentiation, ASCs were exposed to a supernatant from fibroblast cells under black jade extract for a day, and then were exposed to a supernatant from fibroblast cells under benzo(a)pyrene.

### 2.2. Quantitative PCR

Total RNA in cells was extracted using a RiboEx reagent (GeneAll, Seoul, Korea). The RNA was then reverse transcribed into cDNA using a Maxime RT PreMix (iNtRON, Seongnam, Korea) and quantitative PCR was performed with primers (Table 1) at the cycling parameters: 1 min at 95 °C, followed by 35 cycles of 35 s at 59 °C and 1 min at 72 °C for 35 cycles. Amounts of amplified target genes in samples were compared with the house keeping gene, GAPDH, and relative quantities of target genes were determined using the quantities of the control.

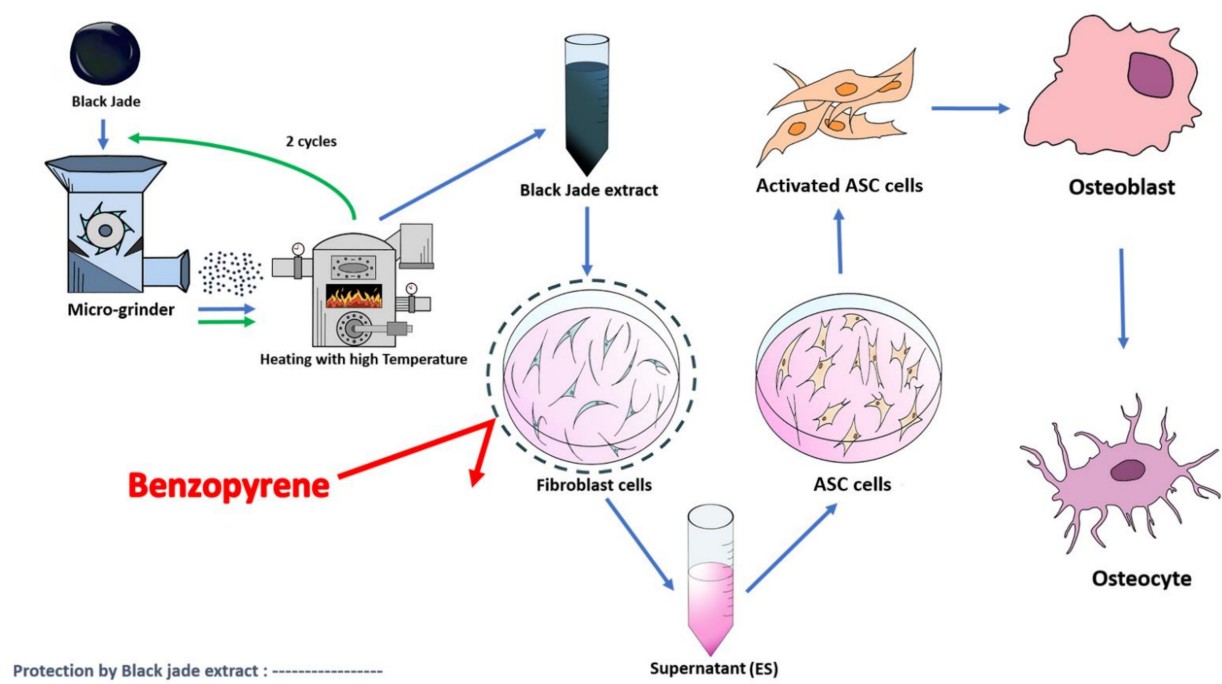

**Figure 1.** A schematic flow chart for the research procedures. ASC: adipose-derived stem cell.

**Table 1.** The list of primers for qRT-PCR.

| Primer | F/R * | Seq (5′ → 3′) |
|---|---|---|
| AKT | F | GGCTGCCAAGTGTCAAATCC |
| | R | AGTGCTCCCCCACTTACTTG |
| NFκB-P50 | F | CGGAGCCCTCTTTCACAGTT |
| | R | TTCAGCTTAGGAGCGAAGGC |
| NFκB-P52 | F | AGGTGCTGTAGCGGGATTTC |
| | R | AGAGGCACTGTATAGGGCAG |
| Bcl2 | F | CTGCTGACATGCTTGGAAAA |
| | R | ATTGGGCTACCCCAGCAATG |
| BAX | F | AGCGCTCCCCCACTTACTTG |
| | R | GACAGGGACATCAGTCGCTT |
| Cyt | F | ATGAATGACCACTCTAGCCA |
| | R | ATAGAAACAGCCAGGACCGC |
| Dlx5 | F | ACCATCCGTCTCAGGAATCG |
| | R | ACCTTCTCTGTAATGCGGCC |
| Runx2 | F | GACCAGTCTTACCCCTCCTACC |
| | R | CTGCCTGGCTCTTCTTACTGAG |
| VEGF | F | ACTGCCATCCAATCGAGACC |
| | R | CGGCCGCGGTGTGTCTA |
| BMP6 | F | ATCCTTTCTGCGAGCGGGTT |
| | R | ATCTCTCATGGTCGTCCGGG |
| SAMD4 | F | AGCATGGGGTGTGAGAATGG |
| | R | TCGTTTCAAAGGGCTGTGGT |
| COL1A1 | F | CAGGGTGGCTTCTGATATGTCC |
| | R | GGTTAGAAAGTGGCAAAGGGGA |
| GAPDH | F | GTGGTCTCCTCTGACTTCAACA |
| | R | CTCTTCCTCTTGTGCTCTTGCT |

### 2.3. Flow Cytometry

To analyze the osteogenic differentiation in ASCs, all cellular samples were fixed with 2% paraformaldehyde for 4 h and treated with 0.02% Tween 20 for 5 min. After blocking using an Fc blocking solution (BD science) [17], samples were treated with Fluorescein isothiocyanate (FITC)-anti-osteocalcin (Biotechne, Minneapolis, MN, USA), PE-anti-osteoponin (Biotechne), Alexa Fluor 647-anti-SOST/sclerostin (Novus, Centennial, CO, USA) for 2 days. These treated samples were washed using phosphate buffered saline (PBS) and analyzed using a flow cytometer (BD FACScalibur) and FlowJo 10.7.0 (BD science) [18]. To estimate the viability, the dermal cells were stained with Annexin V-PI (Invitrogen) and measured using a flow cytometer (BD FACScalibur) and FlowJo 10.6.1 (BD biosciences).

### 2.4. Image Analysis

All samples were fixed with 2% paraformaldehyde for 4 h and treated with 0.02% Tween 20 for 5 min. After blocking with 10% bovine serum albumin (BSA) (Sigma) and Fc blocking solution (BD science, San Jose, CA, USA), samples were treated with the three fluorescence-labeled antibodies, including FITC-anti-osteocalcin (Biotechne, Minneapolis, MN, USA), PE-anti-osteoponin (Biotechne), Alexa Fluor 647-anti-SOST/sclerostin (Novus, Centennial, CO, USA) for 2 days. For visualization of calcium, the fixed ASCs were stained with Alizarin Red S (Sigma) for 45 min. After staining, the osteogenic differentiation was analyzed using a fluorescence microscope (Eclipse Ts-2, Nikon, Shinagawa, Japan) and imaging software, NIS-elements V5.11 (Nikon).

### 2.5. Statistical Analysis

All experiments were performed in triplicate ($n$ = 3), and the data were analyzed by a one-way analysis of variance (ANOVA) with the Post Hoc test (Scheffe's method), independent $t$ test using SPSS software v26 (IBM, New York, NY, USA) and Prism 7 (GrapgPad, San Diego, CA, USA) softwares.

## 3. Results and Discussion

The goals in this research were to document the prevention of the extract for apoptosis of fibroblast cells under benzo(a)pyrene secreted substances in their supernatants. Additionally, we supposed that the extract and benzo(a)pyrene inducing supernatants (ES and BPS) from fibroblast cells are effective in the osteogenic differentiation of ASCs localized in subcutaneous tissue.

Black jade extract was used to treat fibroblast cells, and the activated fibroblast cells were exposed to benzo(a)pyrene, one of atmospheric fine particles. ASCs in the conditioned media were analyzed for osteogenic differentiation (Figure 1). It has been reported that overexposure to manganese and chromium induces severe side effects, including neurological dysfunctions, dermatitis, eczematous skin reactions, inflammation, and mitochondrial oxidative stress [19,20]. For maximum effect on the dermal cells, black jade was modified through two cycles of fragmentation and heating (Figure 1).

Black jade extract was used to treat fibroblasts and the resulted supernatant was used to treat adipose-derived stem cells (ASCs) (Figure 1). The activated ASCs differentiated to osteocytes. Fibroblast cells were exposed to the extract to estimate the protective effect of apoptosis of the cells under benzo(a)pyrene. The supernatant from fibroblast cells under the extract were exposed to ASCs to estimate osteogenic differentiation of the cells.

Jade is known as a gemstone, and has a different mineral composition than nephrite and jadeite [1,2]. Compared to the control, the rates of oxidized substances in the heated black jade changed, with increasing oxidized substances without $Al_2O_3$, particularly for magnesium oxide, silicon dioxide, and aluminum oxide (Table 2). Unlike the compositions of the original black jade from Korea, the heated black jade compositions changed significantly, decreased in Mn, Cr, $TiO_2$, and $Al_2O_3$, but increased in FeO, $K_2O$, and $Na_2O$ (Table 2).

**Table 2.** Comparison of compositions and contents between the black jade and the heated black jade.

| Composition | Control * (%) | Heated Black Jade (%) |
|---|---|---|
| $SiO_2$ | 41.30 ± 0.1 | 41.80 ± 0.2 [#] |
| MgO | 34.20 ± 0.45 | 34.80 ± 0.1 [#] |
| Total Fe | 6.78 ± 0.1 | 6.62 ± 0.2 [#] |
| $Al_2O_3$ | 2.81 ± 0.2 | 1.32 ± 0.2 [#] |
| CaO | 1.57 ± 0.3 | 1.68 ± 0.1 [#] |
| FeO | 0.59 ± 0.02 | 0.98 ± 0.01 [#] |
| $K_2O$ | 0.42 ± 0.02 | 0.62 ± 0.02 [#] |
| $Na_2O$ | 0.38 ± 0.01 | 0.74 ± 0.01 [#] |
| $TiO_2$ | 0.16 ± 0.02 | 0.025 ± 0.0002 [#] |
| Cr | 0.07 ± 0.01 | <0.005 ± 0.0001 [#] |
| Mn | 0.08 ± 0.001 | <0.005 ± 0.0002 [#] |

* The control is normal black jade without heating. [#] Significant difference ($p < 0.05$, $n = 3$), independent $t$ test.

To enhance bioactivity, black jade was grinded for two cycles. Based on the results, the peak size in the first cycle was 4.42 μm, which is approximately five times lower than that in the control (Figure 2). In addition, the peak size in the second cycle was approximately 1.22 times lower than that in the first cycle, and the content of particles under 10 μm was 98.83% (Figure 2). Plasma membrane vesicles were approximately 0.1 and 1 μm in diameter. Half of the particles in the black jade in the second cycle were below 3.61 μm (Figure 2). These results suggest that this preparation of black jade extract could attenuate the cytotoxicity and side effects of black jade usage. Moreover, the modified black jade is physically more effective for human cells.

Based on the levels of survival markers, black jade extract activated the expression of anti-apoptotic proteins including AKT, NFκB-P50/52, and Bcl2, and protected fibroblast cells against benzo(a)pyrene (Figure 3a, Supplementary File S1). All anti-apoptotic markers were upregulated in the activated fibroblast cells using the extract, with an approximate 1.7 times increase (Figure 3a). Notably, based on black jade's protective effect, the anti-apoptotic markers were upregulated in activated fibroblast cells exposed to benzo(a)pyrene with an approximate 1.6 times increase (Figure 3a, Supplementary Materials S1). More-over, using the black jade extract, the expression of BAX and Cyt in fibroblast cells was suppressed. In particular, BAX expression was suppressed 2.8 times compared to that in cells exposed to benzo(a)pyrene alone (Figure 3). Although fibroblast cells were exposed to benzo(a)pyrene, the black jade extract upregulated anti-apoptotic markers and suppressed the expression of the apoptotic markers in the activated fibroblast cells (Figure 3a, Supplementary Materials S1). In sharp contrast, without the extract, the fibroblast cells showed downregulated anti-apoptotic markers besides increased apoptotic markers under benzo(a)pyrene (Figure 3a, Supplementary Materials S1). Unlike the cellular viability, which was attenuated about 8.3 times more than the control under benzo(a)pyrene, the extract enhanced cellular viability about 2.9 times more than the exposure to benzo(a)pyrene (Figure 3b). Based on reports from the World Health Organization (WHO), fine particles could induce several diseases including lung cancer, cardiovascular diseases, respiratory diseases, skin inflammation, and allergy [21]. Notably, skin is directly affected by atmospheric fine particles leading to dermatological problems, including dermatitis and dermal irritations [22]. In addition, fibroblast cells under stress upregulate the release of pro-inflammatory cytokines and secrete detrimental signals to various peripheral cells types including immune cells, adipocytes, and stem cells [22]. Overall, the black jade extract protects dermal cells against oxidative stress induced by benzo(a)pyrene.

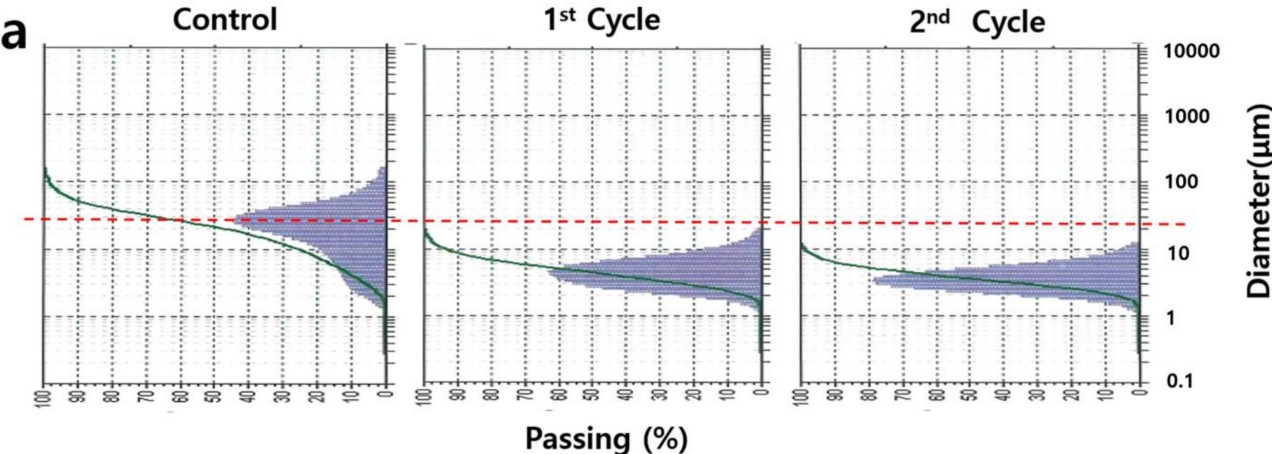

**Figure 2.** Diameters of particles in the grinded black jade. Distributions of particle size in one and two cycles of heating and grinding. (**a**) Red dotted lines indicate the peaks for the particle size of the greatest portion in the black jade particles. (**b**) Comparison of peak sizes between first and second cycles ($p < 0.05$, $n = 3$).

| Samples | Peak Size (μm) | % (< 10 μm) |
|---|---|---|
| Control | 21.35 | 25.74 |
| 1st Cycle | 4.42 | 93.87 |
| 2nd Cycle | 3.61 | 98.83 |

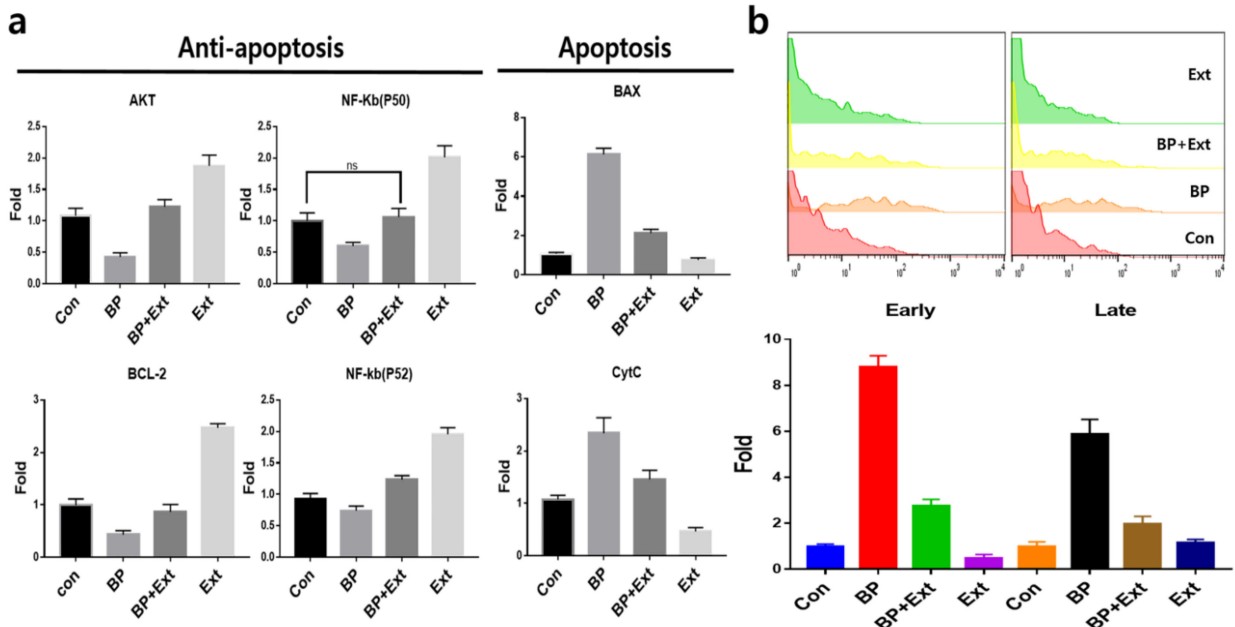

**Figure 3.** Expression levels for anti-apoptotic and apoptotic markers in fibroblast cells exposed to black jade extract. (**a**) The bar graphs indicating relative fold changes for the markers. (**b**) Histograms for the cellular viability under benzo(a)pyrene (BP) or the black jade extract (Ext). BP+Ext represents that fibroblast cells were exposed to BP after treatment using Ext. Con; control, ($p < 0.05$, $n = 3$).

Figure 4, Supplementary Materials S1 and S2 show the expression of osteogenic genes in ASCs exposed to conditioned media (the supernatants of fibroblast cells exposed to

several conditions: control, black jade extract and benzo(a)pyrene). Unlike ASCs in the medium conditioned with benzo(a)pyrene (BPS) that showed downregulated osteogenic genes, the medium conditioned with black jade extract (ES) showed activated expression of pre-osteoblatic genes. Under BPS, all markers were downregulated and notably, BMP6 and COL1A1 were strongly decreased in ASCs (Figure 4). However, all markers in ASCs were strongly upregulated under BPS+ESm (Figure 4). Comparing between BPS and BPS+ES conditioned media, the levels of Runx2 and Dlx5 were 3.89 and 2.40 times higher, respectively, in the latter (BPS+ES) (Figure 4). Interestingly, although the conditioned medium (BPS) suppressed osteoblastic differentiation of ASCs, BPS+ES and ES conditioned media showed the opposite effect (Figure 5a,b). Osteoblastic differentiation markers osteoponin and osteocalcin were upregulated in BPS+ES, along with the highest cell counts at day 10 (Figure 5a,b). At day 10, osteoponin and osteocalcin levels in the BPS+ES conditioned medium were 1.84 and 2.63 times higher, respectively, than those in the BPS conditioned medium (Figure 5c). The conditioned media of activated fibroblast cells enhanced the osteogenic differentiation of the ASCs. With BPS, osteogenic differentiation was attenuated significantly in ASCs (Figure 4). However, after exposure to ES, ASCs activated their osteogenic differentiation under benzopyrene (Figures 4 and 5). Based on several reports, Notch signaling controls osteogenic differentiation of ASCs as well as osteogenic development [23–25]. Although Notch protein suppresses osteogenic differentiation from mesenchymal stem cells by downregulating Wnt/β-catenin and Runx2, Notch activates mineralization during osteogenic development [25]. Bioactive components from black jade in the media-affected Notch-related osteogenic pathways reduced the suppression of osteogenic differentiation in dermal cells, and activated the differentiation besides the protection of dermal cells against fine dust.

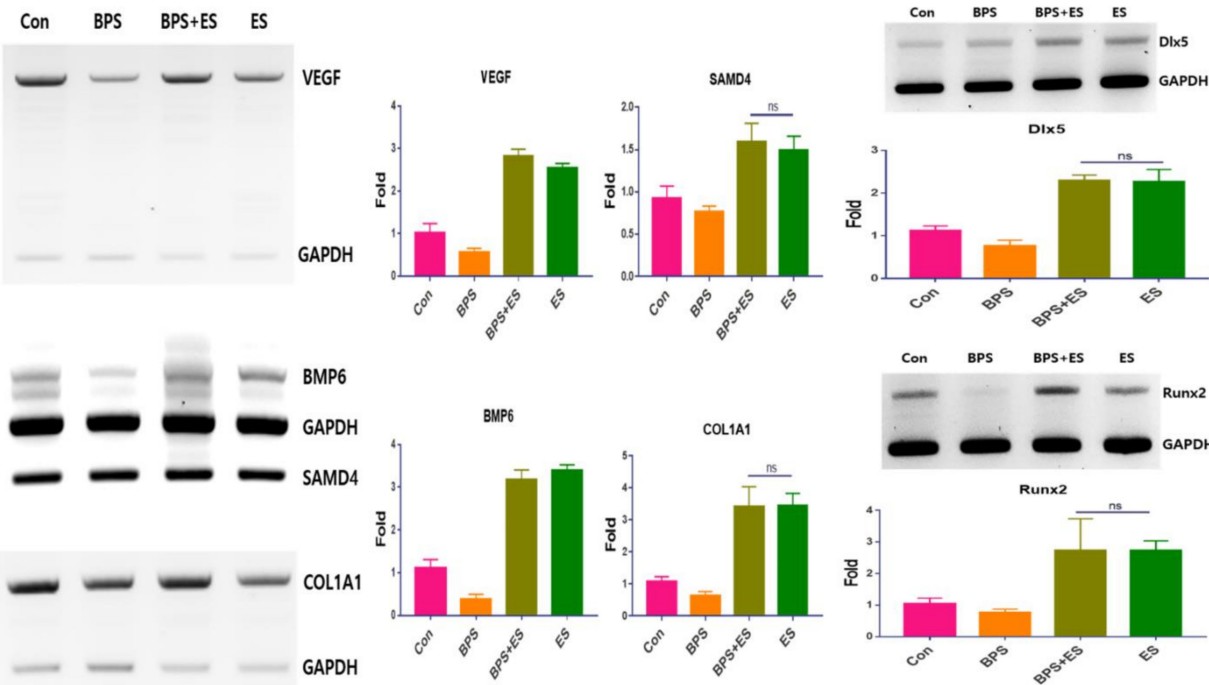

**Figure 4.** Expression of osteogenic genes in adipose-derived stem cells (ASCs) exposed to conditioned media. Bands indicating cDNA levels for the osteogenic markers. Histograms indicating relative fold changes for the corresponding intensity levels. BPS+Ext represents that ASCs were exposed to the conditioned medium after treatment using black jade extract (Ext). The conditioned medium (BPS) contained the supernatant from the fibroblast cells exposed to benzo(a)pyrene. The conditioned medium (ES) contained the supernatant from the fibroblast cells exposed to the extract. Con; Control, ($p < 0.05$, $n = 3$). BPS: benzo(a)pyrene inducing supernatants; ES: extract supernatants.

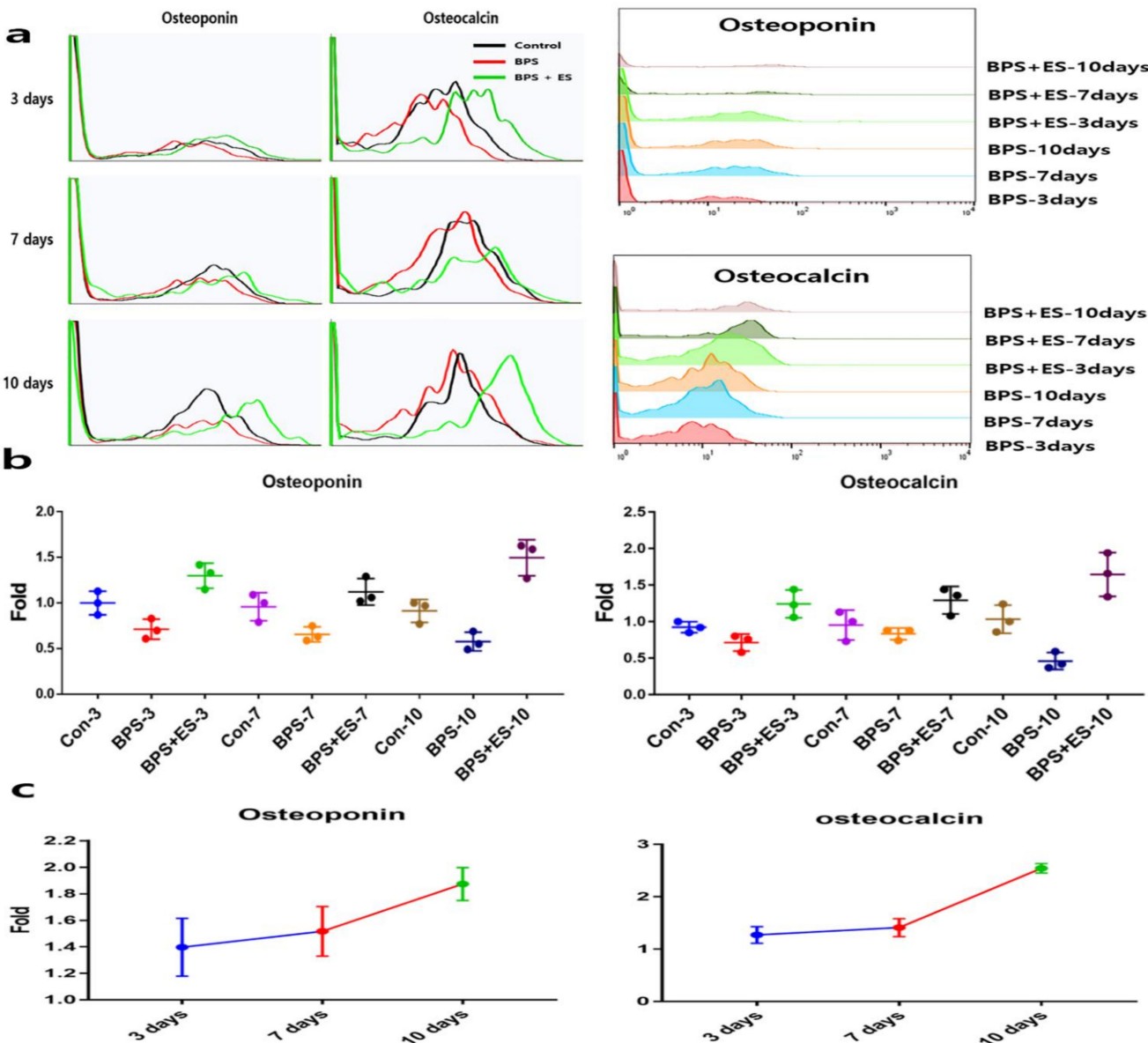

**Figure 5.** Protective effect of black jade extract on osteoblastic differentiation in adipose-derived stem cells (ASCs) under conditioned mediums. These results are from ASCs under conditioned mediums (BPS, BPS+EX). (**a**) Histograms indicating cell counts and expression levels for the osteoponin and osteocalcin detected using flow cytometry. (**b**) Relative fold changes for cell counts in the corresponding graphs. (**c**) Relative fold changes between BPS and BPS+ES. After exposure to ES, ASCs were exposed to BPS. The conditioned medium (BPS) contained the supernatant from the fibroblast cells exposed to benzo(a)pyrene. The conditioned medium (ES) contained the supernatant from the fibroblast cells exposed to the extract. Con; Control, ($p < 0.05$, $n = 3$).

Unlike BPS inhibited osteoblast differentiation, ES protected and activated osteoblastic differentiation of ASCs (Figure 6a,b and Figure 7). Cell counts for osteocytes were about 3.9 times higher than those in BPS at day 10 (Figure 6b). Moreover, osteoblastic differentiation was gradually increased during the 10 days in BPS+ES (Figure 6c). Despite being under BPS, ASCs exposed to ES strongly activated osteogenic differentiation (Figure 7). Although sclerostin secreted by osteoclasts inhibits osteoblastic maturation and bone formation, the protein is an important marker to differentiate between the osteoblast and osteocyte during early osteogenesis development [26]. The upregulation of sclerostin has been reported in osteocytes surrounding bone, osteoclasts under stresses, developing bone and bone marrow-derived osteoclast precursors, to decrease the formation of osteoclasts [26]. These

results suggest that the extract has the potential to modulate the formation of osteoclasts during bone development.

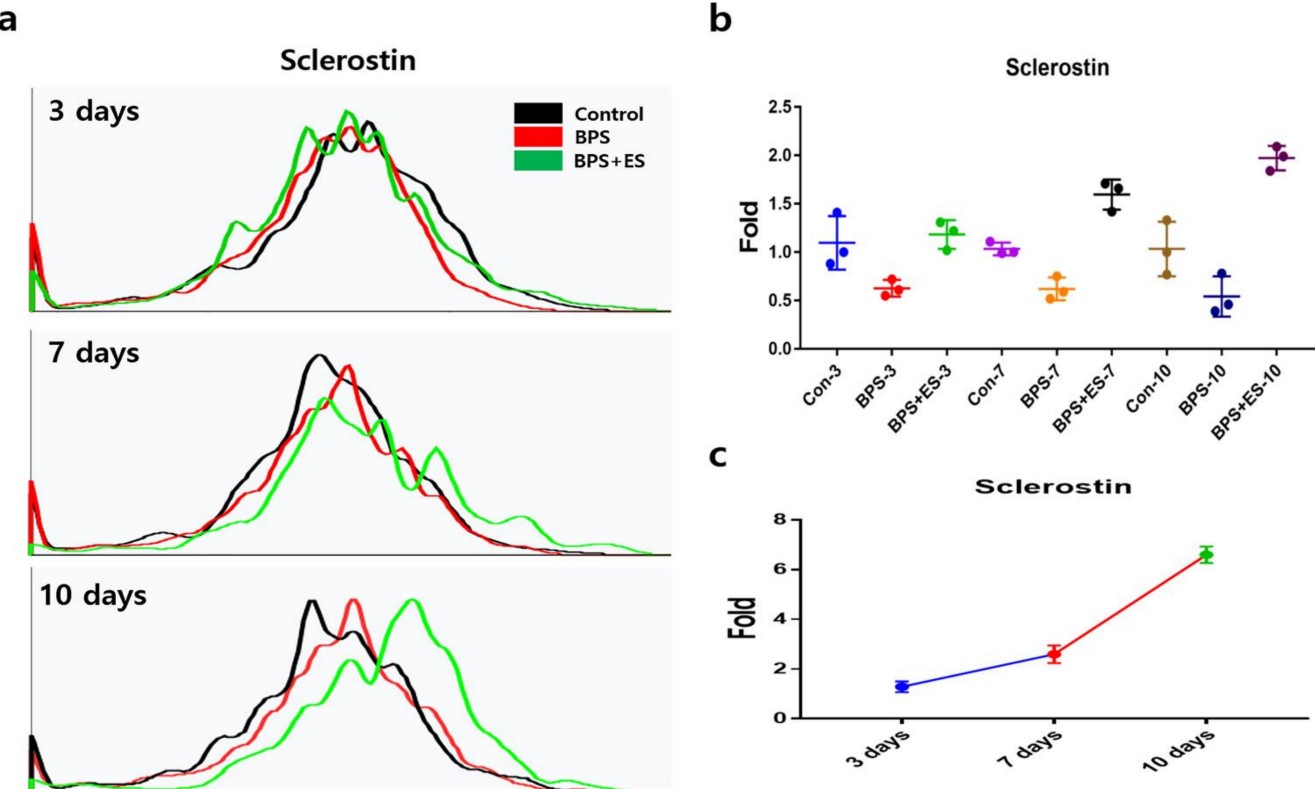

**Figure 6.** Protective effect on osteoblast differentiation by black jade extract in adipose-derived stem cells (ASCs) under conditioned mediums. These results are from ASCs under conditioned mediums (BPS, BPS+EX). (**a**) Graphs indicating sclerostin cell counts detected using flow cytometry. (**b**) Relative fold changes for sclerostin cell counts. (**c**) Relative fold changes between BPS and BPS+ES. BaP+ES means that ASCs were exposed to BPS after treatment using ES. The conditioned medium (BPS) contained the supernatant from the fibroblast cells exposed to benzo(a)pyrene. The conditioned medium (ES) contained the supernatant from the fibroblast cells exposed to the extract. Con; Control, ($p < 0.05$, $n = 3$).

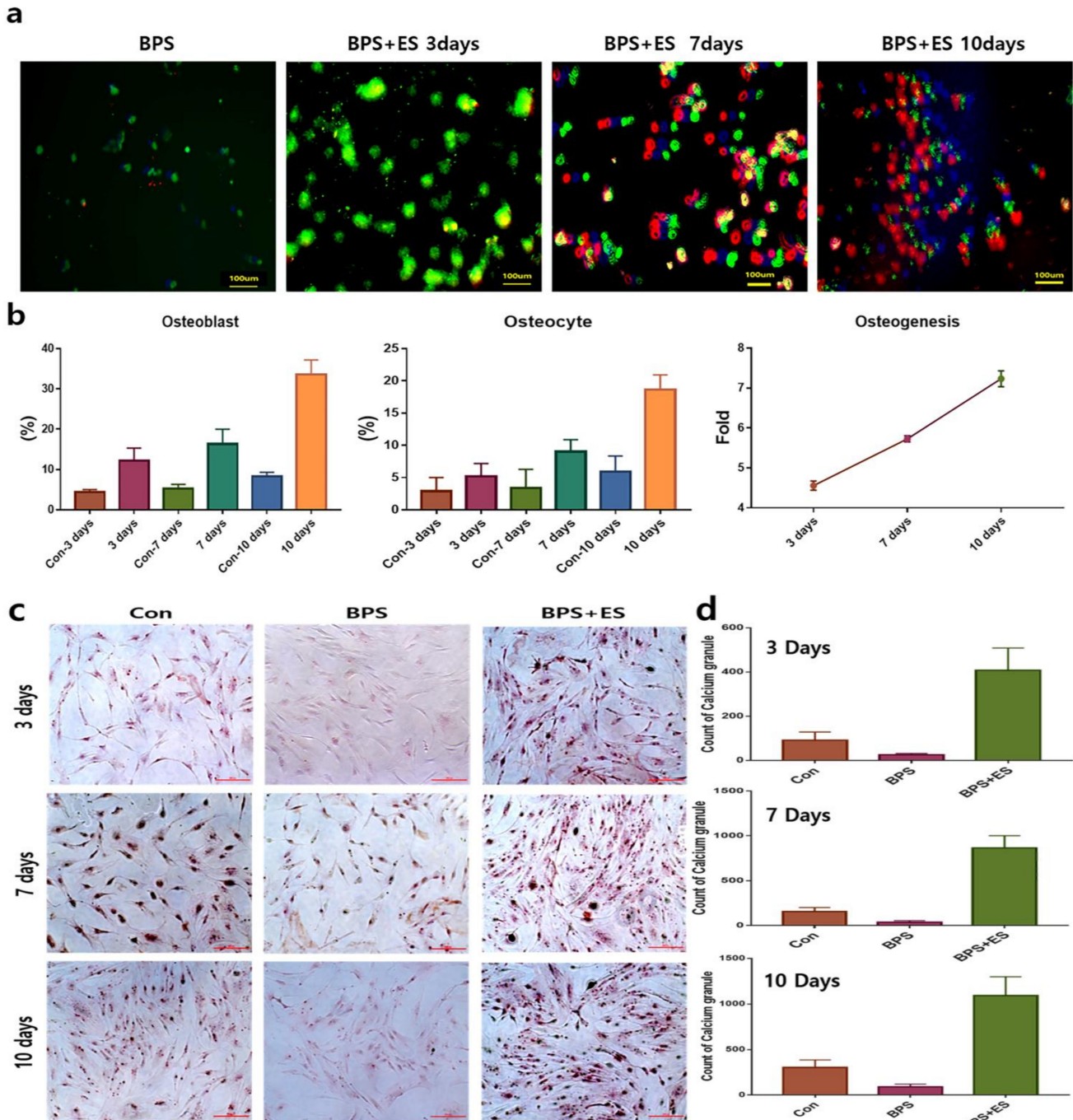

**Figure 7.** Image analysis for differentiated osteoblasts and osteocytes of adipose-derived stem cells (ASCs) cultured in the conditioned media. (**a**) Images indicating stained cells for osteoponin, osteocalcin, and sclerostin, detected using fluorescence-labeled antibodies and a microscope. Green; osteocalcin, blue: osteoponin, red; sclerostin. (**b**) Relative fold changes for the corresponding stained cell counts. (**c**) ASCs staining by using alizarin Red S stain under various conditions. (**d**) Results from the counting of calcium granules in ASCs using software, NIS-Elemnets V5.11. Relative fold changes between BPS and BPS+ES. BPS+ES represents that ASCs were exposed to BPS after treatment using ES. The conditioned medium (BPS) contained the supernatant from the fibroblast cells exposed to benzo(a)pyrene. The conditioned medium (ES) contained the supernatant from the fibroblast cells exposed to the extract. Con; Control, ($p < 0.05$, $n = 3$).

## 4. Conclusions

The black jade extract protects dermal cells against air pollution and benzo(a)pyrene, and upregulates the secretion of bioactive substances from dermal cells to enhance the osteogenic differentiation of ASCs. In addition, air pollution aggravates the inhibition of

osteogenic differentiation of ASCs besides the apoptosis of dermal cells. Therefore, black jade extract is very useful when applied as a functional biomaterial.

**Supplementary Materials:** The following are available online at https://www.mdpi.com/2076-3417/11/3/1346/s1, File S1: full gels for qPCR, File S2; concentration of IL-1 and TNF.

**Author Contributions:** Conceptualization and methodology, Y.P., G.H.S.; writing–original draft preparation, Y.P.; writing–review and editing, B.K.; supervision, B.K. and S.G.L.; funding acquisition, G.S.J. and S.J. All authors have read and agreed to the published version of the manuscript.

**Funding:** This research received no external funding.

**Institutional Review Board Statement:** Not applicable.

**Informed Consent Statement:** Not applicable.

**Data Availability Statement:** Not applicable.

**Acknowledgments:** This study was supported by Dr. JADE Co., Ltd., LIFE TOGETER and Korea University.

**Conflicts of Interest:** The authors declare no conflict of interest.

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
