# Peer review of "Effects of Black Jade on Osteogenic Differentiation of Adipose Derived Stem Cells under Benzopyrene"

_applsci, doi:10.3390/app11031346_

Round 1

Reviewer 1 Report

This is an interesting paper that investigated the effects of black jade for skin under benzopyrene. I recommend to publish subject to the following minor revisions:

  1. In page 3, line 87, what does PCR stand for? Add a sentence to explain what does Quantitative PCR measure?
  2. In page 3, line 95, add a sentence to explain what does Flowcytometry measure?
  3. Figure 5a needs to have higher resolution.
  4. In Figure 7a, what do red, green, and blue color represent? is it: osteoponin, osteocalcin, and sclerostin?

Author Response

I appreciated for your excellent comments for my manuscript. Our manuscript has been considerably improved through your comments. I have attached the file containing answers for your comments.  

Reviewer 2 Report

Major concerns:

  • As an additional condition, the control compound should be used in conjunction with benzopyrene (i.e., BP+Con condition) for evaluating the effect of black jade extract versus black jade control when challenged with benzopyrene.
  • A more relevant treatment plan would be to treat cells first with benzopyrene, then to the black jade extract if the goal is to determine whether this extract can reverse cellular damage caused by benzopyrene.
  • Since apoptotic cells are well-known to release different factors, it is unclear whether changes in ASC differentiation between BPS and BPS+ES conditions (Figs. 4-7) are caused directly by the different treatments on fibroblast cells, or as a result of the treatment’s effect on fibroblast cell apoptosis that is described in Figure 3.
  • The link between damage and the inhibition of osteogenic differentiation, and its relevance in skin, is unclear as written. Please clarify why it is relevant to examine osteogenic differentiation of ASCs as a marker of the skin’s resistance to damage.

Minor concerns:

  • The word “skin” should be removed from the title, since skin was not evaluated in vivo.
  • The title is vague as written and would be improved by including a more descriptive article title that describes the major findings.
  • Concluding that “black jade is an excellent material to be utilized as a functional component in cosmetic and pharmacy” on line 27 should be tempered, as the current study did not evaluate the safety or efficacy of its use in vivo.
  • Indicate which differences are significant in Table 1.
  • Figure 1: (1) “ADSC” was previously abbreviated “ASC” in text. Make consistent. (2) “Benzopyrene” is confusing as illustrated. Convention suggests by the figure that fibroblast cells inhibit benzopyrene, when benzopyrene is being used as a treatment following black jade extract.
  • Clarify how the control compound was generated in the material and methods.
  • Line 21: change the word “reversed.” Since benzopyrene treatment follows black jade, the effect is not reversed as written. Perhaps “counter-acted” would be more appropriate.
  • Line 196: include data or cite relevant literature to suggest that this preparation of black jade extract “could attenuate the cytotoxicity and side effects” of black jade usage.
  • Lines 283-284: temper conclusion to reflect in vitro findings, since "the bone microenvironment during bone development" was not assessed.
  • Indicate where significant differences were detected in data throughout or include ANOVA tables.
  • Describe and refer to the supplemental material in the manuscript text.
  • Discuss Figure 7 results in the manuscript text. This is missing as written.
  • Fig 7A: Is this immunostaining? This should be clarified in the legend along with what the colors correspond to in panel A.
  • Explain the relevance of treating ASCs with supernatant from treated fibroblasts rather than treating ASCs directly.
  • The titles of Figures 5 and 6 do not mention conditioned medium and suggest that ASCs were treated directly. Please clarify.
  • Clarify if there is a difference between BP and BPS or make terms consistent.
  • Clarify if there is a difference between ES and ext or make terms consistent.

Author Response

(The authors gave the same response as above.)

Round 2

Reviewer 2 Report

The manuscript is improved. My minor comments are included below.

  1. Please include which statistics were applied in Table 1. The "statistical analysis" section in the methods suggest ANOVA was used for all analyses, but in the case of Table 1, only two groups are being compared so ANOVA is not appropriate. 
  2. The manuscript needs careful proofing for spelling and grammatical errors. (For example: "between" misspelled line 67, "fibroblast" misspelled line 88, "flow cytometry" throughout should be two words, "significant" misspelled line 171, the word "supported" on line 80 should be "supplied", "Thus" on line 67 should be "this")
  3. The last sentence of the introduction (lines 68-70) should be revised to reflect the results of your current study. The result of your study does not directly contribute to the development of functional cosmetic goods that are listed. This speculation should be tempered, and it is perhaps better suited for the conclusion section.

Author Response

  1. Please include which statistics were applied in Table 1. The "statistical analysis" section in the methods suggest ANOVA was used for all analyses, but in the case of Table 1, only two groups are being compared so ANOVA is not appropriate. 

Answer: The results of Table1 were analyzed using independent T test.

We added these texts, “independent T test” at line 120 and 168

  1. The manuscript needs careful proofing for spelling and grammatical errors. (For example: "between" misspelled line 67, "fibroblast" misspelled line 88, "flow cytometry" throughout should be two words, "significant" misspelled line 171, the word "supported" on line 80 should be "supplied", "Thus" on line 67 should be "this")

Answer: We revised misspelled words in the manuscript besides the referred words

  1. The last sentence of the introduction (lines 68-70) should be revised to reflect the results of your current study. The result of your study does not directly contribute to the development of functional cosmetic goods that are listed. This speculation should be tempered, and it is perhaps better suited for the conclusion section.

Answer: We deleted the sentence from Introduction because there are the similar sentence in Conclusion.